# Approaches of the Innate Immune System to Ameliorate Adaptive Immunotherapy for B-Cell Non-Hodgkin Lymphoma in Their Microenvironment

**DOI:** 10.3390/cancers14010141

**Published:** 2021-12-28

**Authors:** Takashi Watanabe

**Affiliations:** Department of Personalized Cancer Immunotherapy, Mie University Graduate School of Medicine, 2-174, Edobashi, Tsu City 514-8507, Japan; twatanabe@med.mie-u.ac.jp

**Keywords:** CD47, CpG oligodeoxynucleotide, 4-1BB, OX40, STING, innate immunity, diffuse large B-cell lymphoma, follicular lymphoma, mantle cell lymphoma

## Abstract

**Simple Summary:**

The innate immune checkpoint blockade anti-CD47 antibodies combined with the anti-CD20 antibody rituximab can invigorate macrophage phagocytosis. Generally applied as vaccine adjuvants, TLR9 agonist CpG-oligodeoxynucleotides (CpG-ODNs) stimulate antigen presenting cells (APCs) and plasmacytoid dendritic cells (pDCs) to secrete interferon (IFN)-α/β, which activates natural killer (NK) cells. This innate immune activation and pDC maturation leads to potentiation of the adaptive immune response. In fact, stimulator of interferon gene agonists also induce APCs to secrete IFN-α/β. CpG-ODNs, combined with rituximab or irradiation, demonstrated clinical efficacy in indolent B-cell non-Hodgkin lymphoma patients. Furthermore, patients with mantle cell lymphoma presenting Ig-derived neoantigens and vaccinated with CpG-ODN priming T cells maintained minimal residual disease. Additionally, IFN-α–conditioned DC vaccination as well as 4-1BB and OX40 agonist antibodies activate NK cells as well as T cells. Collectively, these innate activators are promising agents for enhancing anti-B-cell-lymphoma adaptive immunity, such as chimeric antigen receptor-T-cell therapy.

**Abstract:**

A dominant paradigm being developed in immunotherapy for hematologic malignancies is of adaptive immunotherapy that involves chimeric antigen receptor (CAR) T cells and bispecific T-cell engagers. CAR T-cell therapy has yielded results that surpass those of the existing salvage immunochemotherapy for patients with relapsed/refractory diffuse large B-cell lymphoma (DLBCL) after first-line immunochemotherapy, while offering a therapeutic option for patients with follicular lymphoma (FL) and mantle cell lymphoma (MCL). However, the role of the innate immune system has been shown to prolong CAR T-cell persistence. Cluster of differentiation (CD) 47-blocking antibodies, which are a promising therapeutic armamentarium for DLBCL, are novel innate immune checkpoint inhibitors that allow macrophages to phagocytose tumor cells. Intratumoral Toll-like receptor 9 agonist CpG oligodeoxynucleotide plays a pivotal role in FL, and vaccination may be required in MCL. Additionally, local stimulator of interferon gene agonists, which induce a systemic anti-lymphoma CD8^+^ T-cell response, and the costimulatory molecule 4-1BB/CD137 or OX40/CD134 agonistic antibodies represent attractive agents for dendritic cell activations, which subsequently, facilitates initiation of productive T-cell priming and NK cells. This review describes the exploitation of approaches that trigger innate immune activation for adaptive immune cells to operate maximally in the tumor microenvironment of these lymphomas.

## 1. Introduction

Since the introduction of rituximab plus cyclophosphamide, doxorubicin, vincristine, and prednisone (R-CHOP) in 1997, 60–65% of patients with diffuse large B-cell lymphoma (DLBCL) have been cured and the ratio of relapsed patients after achieving complete response (CR) is 20–25% and that of refractory cases is approximately 10–15% [1]. For relapsed or refractory (R/R) follicular lymphoma (FL) patients, new therapies, such as the immunomodulatory drug lenalidomide combined with the anti-CD20 monoclonal antibody (MoAb) rituximab [2] and the selective reversible inhibitor of the p110δ isoform of phosphoinositide 3-kinase inhibitor idelalisib [3], has been approved. As for patients with R/R mantle cell lymphoma (MCL), the irreversible Bruton’s tyrosine kinase (BTK), which is a critical member of the B-cell receptor signaling pathway, the inhibitor ibrutinib has been approved [4]. In addition to directly affecting the neoplastic B cells, idelalisib impairs the expansion of follicular helper T cells, which support FL growth [5], and ibrutinib shifts the balance of T cell responses toward the more therapeutically effective Th1 subset in their microenvironment [6]. However, follicular lymphoma (FL) patients who respond to currently available therapies can only be cured by allogenic transplantation if they are relatively young, especially those who experience disease progression within 24 months after receiving first-line immunochemotherapy. Despite this, the needs of 15–20% of FL patients are still unmet [7,8,9,10], although FL patients have a long survival [11] and could be successfully managed with new drugs including the enhancer of zeste homolog 2 inhibitor tazemetostat [12]. Moreover, mantle cell lymphoma (MCL) patients cannot maintain remission even after high dose immunochemotherapy followed by autologous stem cell transplantation (ASCT) [13]. Ibrutinib has improved progression-free survival (PFS) [4] and rituximab maintenance prolonged duration of response in relapsed or refractory (R/R) MCL patients [14].

For these patients, chimeric antibody receptor (CAR) T-cell therapy has emerged as a promising treatment [15,16,17,18,19], and long-term, durable remission was obtained in 30–50% of patients with DLBCL [17,20,21,22]. Approaches to immune checkpoint blockades after relapse following CAR-T therapy only show temporary responses [23]. The aforementioned population cannot survive long after relapse [24]. CAR T-cell dose was associated with development of severe (Grade (G) ≥ 3) cytokine release syndrome [25] and peak CAR T-cell expansion was significantly related to severe (G ≥ 3) immune effector cell-associated neurotoxicity syndrome [26].

The phase 3 trial of idiotype vaccination following remission achieved after chemotherapy failed to show clinical benefit for FL patients compared with keyhole limpet hemocyanin (KLH) alone, with the adjuvant granulocyte macrophage colony-stimulating factor (GM-CSF) [27]. Instead, as for MCL patients, the immunomodulatory drug lenalidomide has been proven to be effective [28]. It is important to realize the concept of an immunity cycle [29]. The maximal operation of adaptive immune cells in the tumor microenvironment is a prerequisite for activating the innate immune system.

The innate immune response involves the activation of a variety of cell types, such as dendritic cells (DCs), macrophages, monocytes, neutrophils, basophils, eosinophils, lymphocytes, and natural killer (NK) cells. The innate immune system responds rapidly to pathogens, initiates pathogen clearance, and assists in the healing of impaired tissues. By contrast, the adaptive immune system principally comprises of B cells and T cells. Adaptive immune responses occur more gradually but are highly specific and therefore critical for maintaining immunity via a long-term memory effect. Triggering innate immune activation involves the anti-cluster of differentiation (CD) of 47 antibodies, which can invigorate macrophages by blocking the “don’t eat me” signal in DLBCL [30,31]. Low-dose irradiation combined with intratumoral (IT) injection of the Toll-like receptor (TLR) 9 agonist oligodeoxynucleotides (ODNs), including certain unmethylated CpG motifs (CpG-ODN), stimulate activated cytotoxic T cells in FL [32].

Tumor-associated macrophages (TAMs), can become polarized toward the classically activated macrophage (M1) phenotype or the protumoral M2 phenotype [33,34]. M1 TAMs elicit a Th1 response that can induce tumor cell killing, whereas M2 TAMs are activated by IL-4 and IL-13 produced by Th2 cells, eosinophils, and basophils. GM-CSF, bacterial products, and interferon-γ trigger the M1 subtype, which secrete pro-inflammatory molecules, including interleukin (IL)-1, IL-6, IL-12, IL-23, TNF-α, nitric oxide (NO), C-X-C motif chemokine ligand (CCXCL)9, CXCL10, CXCL11, and reactive oxygen species. Specifically, a protective function has been reported for C-X3-C motif chemokine receptor 1 (CX3CR1)^+^Ly6C^+^ non-classical patrolling mouse monocytes, which depend on the N4a1 transcription factor and patrol microvasculature in steady state conditions. These cells, which infrequently extravasate into tissues and differentiate into macrophages, quickly accumulate in lung metastases within tissues and repress tumor growth in mouse models [35]. In contrast, M2 TAMs produce immunosuppressive cytokines, such as IL-10 and tumor growth factor (TGF)-β as well as anti-inflammatory chemokines: namely, C-C motif chemokine ligand (CCL)17 and CCL22, both of which recruit regulatory T cells [36]. Circulating precursors that are recruited into tumor tissues, where they differentiate into TAMs include conventional inflammatory monocytes and monocytic-myeloid-derived suppressor cells (M-MDSCs). Down-regulation of the transcription factor signal transducer and activator of transcription 3 (STAT3) plays a vital role in the differentiation of M-MDSCs into mature TAM [37]. Inflammatory monocytes, defined as CCR2^+^Ly6C^+^ cells in mice, have been shown to induce TAM accumulation [38]. Phenotypically, mannose receptor C type 1 (CD206), class A scavenger receptor (also known as CD204), and hemoglobin scavenger receptor (CD163) are currently used as M2-associated markers [39,40,41,42]. M2 TAM induce angiogenesis, immunosuppression, and promotion of cancer growth and metastasis.

Interestingly, immunoglobulin neoantigen peptides derived from the unique VDJ rearrangements and somatic mutations are presented by major histocompatibility complex (MHC) class II (MHC-II) in MCL [43]. In one clinical trial, once MCL patients had achievedt remission after ASCT, they were vaccinated with CpG-stimulated autologous tumor cells followed by the adoptive transfer of vaccine-primed lymphocytes [44]. Additionally, the costimulatory molecule 4-1BB/CD137 [45,46] or OX40/CD134 agonistic antibodies [47] activate dendritic cells, leading to the initiation of productive T-cell priming. Furthermore, although only demonstrated in preclinical models, local stimulator of interferon gene agonists (STINGa) induces a systemic anti-lymphoma CD8^+^ T cell response [48]. This review describes the therapeutic approaches involving the exploitation of the innate immune system in different types of lymphoma.

## 2. “Don’t Eat Me” Signal Blockade Anti-CD47 Antibody Is an Innate Immune Checkpoint Inhibitor

### 2.1. Anti-CD47 Antibodies

CD47, which is a ubiquitous member of the immunoglobulin (Ig) superfamily, binds signal regulatory protein alpha (SIRPα), is phosphorylated the immunoreceptor tyrosine-based inhibitory motif (ITIM) [49] (Figure 1, right), and is expressed on myeloid cells including macrophages [50,51,52]. Using “foreign” sheep red blood cells (RBCs) with an anti-CD47 antibody, at the synapse, antibody-opsonized human RBCs showed accumulation of phosphotyrosine, the focal adhesion molecule paxillin [53], F-actin [54], and the major motor isoform non-muscle myosin-IIA (NMM IIA) (Figure 1, left) [55]. SIRPα localizes to the synapse when CD47 is functional (Figure 1, right), suppressing the accumulation of phosphotyrosine and myosin without affecting F-actin. Human CD47 strongly inhibited phagocytosis of THP-1 cells, a human macrophage cell line, by inducing delocalization of NMM IIA and phospho-paxillin. Accordingly, the CD47-SIRPα interaction initiates dephosphorylation by recruiting the Src-homology 2 domain (SH2)-containing protein tyrosine phosphatase-1 (SHP-1) (Figure 1, right) [51,56,57,58] directed at phosphotyrosine in NMMIIA as well as phospho-Fcgamma receptor (FcγR), downstream phospho-Spleen tyrosine kinase (pSyk) [59], and phospho-Casitas B-lineage lymphoma (pCbl) [55].

The synergistic effect of B6H12.2, an anti-CD47 IgG1 antibody, and rituximab was demonstrated through a mechanism combining the Fc receptor (FcR) and FcR-independent stimulation of phagocytosis [60]. 

In pre-clinical experiments using Raji cells as a disseminated non-Hodgkin lymphoma (NHL) model, the humanized IgG4 isotype of CD47-blocking antibody Hu5F9-G4 synergized with rituximab to eliminate Raji cells, resulting in extended survival of mice, although Hu5F9-G4 alone showed only a modest survival benefit compared with that of the control [61]. 

Because of the high degree of homology (99% identity) between human and cynomolgus macaques CD47 [62], cynomolgus macaques are clinically relevant for evaluating CD47-SIRPα-related toxicity. Using this animal model, no serious adverse effects (AEs) were observed in terms of pharmacokinetics and toxicology [61]. In all animals examined, the nadir of anemia occurred approximately 5–7 days after Hu5F9-G4 infusion and was generally related to the dose administered. The anemia spontaneously resolved, returning to baseline levels after approximately two weeks. In the dose escalation study from 0.1 to 300 mg/kg, the animals tolerated all doses well. Moreover, a study was conducted in which a priming dose of either 1 or 3 mg/kg was administered on day 1, followed a week later by weekly maintenance doses of 30 mg/kg for 6 weeks. A pharmacokinetic analysis indicated that exposure to Hu5F9-G4 as measured by maximum plasma concentration and the area under the serum concentration curve in both animals achieved sustained serum levels within or above the potential therapeutic range for the duration of the maintenance dosing period with a prolonged half-life after the final dose [61].

Clinically, patients with R/R NHL, including DLBCL (*n* = 15, 68%) and FL (*n* = 7, 32%) were enrolled in a phase 1b study of Hu5F9-G4 and rituximab [30]. In total, 50% of the patients achieved CR (36%) or partial response (PR). The rates of overall response (OR) and CR among patients with DLBCL were 40% (6/15 patients) and 33% (5/15 patients), respectively, whereas, among patients with FL, the OR and CR rates were 71% (57 patients) and 43% (37 patients), respectively.

In contrast, a multicenter, open-label, first-in-human phase 1 study of TTI-621 was conducted, wherein a recombinant soluble fusion protein composed of the CD47-binding domain of human SIRPα and the Fc region of human IgG1 [63], binds to CD47 in patients with R/R hematologic malignancies [31]. During the expansion study, TTI-621 was combined with rituximab in patients with B-cell NHL, or combined with nivolumab in patients with non-Hodgkin lymphoma. The primary endpoint was the incidence/severity of AEs, and the secondary endpoint was the overall response rate (ORR). Eighteen patients were enrolled in the dose escalation study, and 146 were enrolled in the expansion study, including 35 rituximab combinations and 4 nivolumab combinations. Due to the observation of transient G4 thrombocytopenia, the maximum tolerated dose (MTD) was determined to be 0.2 mg/kg, 0.1 mg/kg was then assessed in combination cohorts. AEs included infusion-related reactions, thrombocytopenia, chills, and fatigue. Thrombocytopenia of ≥G3 observed in 20% of patients was reversible between doses and not related to bleeding. The ORR in all patients was 13%; 29% (2/7) in DLBCL patients, 25% (8/32) in T-cell NHL patients treated with TTI-621 monotherapy, and 21% (5/24) in DLBCL patients treated with TTI-621 and rituximab [31].

### 2.2. SIRPα-Fc-CD40 Ligand Agonist Bridging Macrophage-Mediated Tumor Cell Phagocytosis to Antigen-Presenting Cell (APC) Activation and Antigen Presentation

CD40 ligation by CD40 ligand (CD40L), which is expressed by resting CD8α^+^ DCs, augments the antigen cross-presenting activity of DCs [64,65,66,67,68]. Whereas CD40 ligation promotes a type I interferon (IFN) response through STING activation, STING activation is not a prerequisite for the antitumor immune response (IR) to CD40 stimulation. However, antitumor immunity to CD40 agonists depends on BATF3^+^ DCs and CD8^+^ T cells [69]. Accordingly, the CD40/CD40L axis can bridge innate and adaptive immunity. A CD47/SIRPα blockade provokes antitumor activity by inducing tumor cell phagocytosis by macrophages and activating the cross-presentation of tumor antigens to CD8^+^ T cells by DCs (Figure 2). Both these processes are potentiated by CD40 signaling. Accordingly, Schreiber et al. developed a SIRPα-Fc-CD40L agonist redirected checkpoint fusion protein to address these distinct pathways with a single biological agent SIRPα-Fc-CD40L is a novel, bifacial fusion protein that combines the extracellular domains of SIRPα and CD40L, adjacent to a fundamental Fc domain [70]. SIRPα-Fc-CD40L binds CD47 and CD40 with high affinity and activates CD40 signaling without FcR cross-linking. SIRPα-Fc-CD40L activated a type I IFN response in macrophages and augmented the activity of antibody-dependent cellular phagocytosis (ADCP)-competent directed antibodies both in vitro and in vivo. SIRPα-Fc-CD40L relieves the inhibitory “don’t eat me” signal, while concomitantly lending an “eat me” signal through co-stimulation of CD40 by CD40L. Therefore, SIRPα-Fc-CD40L could bridge the macrophage-mediated phagocytosis of tumor cells to antigen-presenting cell (APC) activation and antigen presentation by two modes: (1) direct APC activation through CD40/CD40L, leading to antigen-specific CD8 stimulation and programmed immune memory, and (2) direct blocking of CD47 inhibition, resulting in enhanced tumor phagocytosis and increased antigen cross-presentation. Consequently, ADCP antibody combinations augmented targeted tumor phagocytosis, antigen cross-presentation, and antitumor response. SIRPα-Fc-CD40L combined with rituximab provoked significantly greater phagocytosis than the anti-CD47 antibody/rituximab combination [70].

Using A20 cells, additive control of tumor growth was observed when mouse (m) SIRP-α-Fc-CD40L was combined with anti-CD20 antibodies as compared with anti-CD20 antibodies alone or mSIRPα-Fc-CD40L alone. Furthermore, in mice implanted with WEH13 tumors, anti-IFN-α receptor 1 antibody significantly diminished the efficacy of mSIRPα-Fc-CD40L both alone and in combination with anti-CD20 antibodies suggesting that the advantage of the combination weas exclusively attributed to a functional type I IFN response. In the cynomolgus macaque model, no evidence of hemolysis or thrombocytopenia after repetitive infusion of human (h) SIRPα-Fc-CD40L at doses up to 40 mg/kg was observed, although there was dose-dependent variation in the total lymphocyte number between before and after each dose. Additionally, following each hSIRPα-Fc-CD40L infusion in cynomolgus macaques, dose-dependent elevations in serum cytokines/chemokines were observed, such as CCL2, CXCL9, CXCL10, IFN-α, IL-6, IL-15, and IL-23, indicating an on-target pharmacodynamic biology [70].

In summary, the CR rates in DLBCL and FL patients treated with the CD47 antibody, Hu5F9-G4, and rituximab were 33% and 43%, respectively, while the ORR in patients with DLBCL who were treated with TTI-621 (SIPRαFc) and rituximab, was 21%. Moreover, a phase 1 study investigating Hu5F9-G4 in combination with the selective BTK inhibitor, acalabrutinib, has been completed for patients with R/R aggressive NHL (NCT03527147). Meanwhile, a phase 1b/2 study examining magrolimab (Hu5F9-G4) in combination with rituximab or rituximab plus gemcitabine and oxaliplatin in patients with R/R B-cell NHL (NCT02953509) is ongoing.

## 3. Other TAM-Based Therapeutic Approach

### 3.1. Colony-Stimulating Factor-1 Receptor (CSF-1R) Inhibitor 

Recently, three-way FL-follicular dendritic cells (FDC)-macrophage crosstalk has been proposed for FL. Co-cultures of FL patients’ cells from lymph node (LN) biopsies with M2 macrophages generated from monocytes of healthy donors exhibited enhanced angiogenesis via vascular endothelial growth factor (VEGF)-A and angiogenin, dissemination, and immunosuppression. FL-FDC crosstalk supports tumor growth and, through CCL2 and colony stimulating factor-1 (CSF-1) secretion, promotes monocyte recruitment, differentiation, and polarization toward the M2-like phenotype. Moreover, CSF-1 was significantly more abundant in the serum from grade 3A FL patients compared with that of grade 1 or 2 FL patients. High CSF-1 receptor (CSF-1R) expression in FL LN biopsies was also significantly associated with inferior OS and increased risk of histological transformation to DLBCL. Meanwhile, CSF-1R inhibition by pexidartinib (PLX3397) preferentially affected M2 macrophage viability and reprogrammed M2 to M1, thereby, disrupting the FL-M2-derived pro-survival crosstalk [71].

Primary MCL cells produce CSF1R and IL-10. Indeed, significantly higher level of these M2-polarizing factors were detected in the plasma of MCL patients compared with age-matched healthy donors (HDs). Moreover, CD163 was found to be overexpressed on the surface of CD14^+^ peripheral blood (PB) monocytes in several MCL samples compared with HDs. Meanwhile, *CSF1* mRNA expression in MCL plasma was significantly higher in MCL LN compared with MCL PB. Furthermore, *CSF1* mRNA was significantly overexpressed in aggressive (blastoid/pleomorphic) MCL subtypes and related to proliferation in tissues), suggesting a relationship between the MCL/macrophages interplay and tumor aggressiveness. The BTK inhibitor ibrutinib abrogated IL-10 and CSF1 production, the latter of which was achieved by targeting the BCR signaling network, in MCL cells, leading to inhibition of CD163^+^ polarization of the ibrutinib-sensitive MCL cells, thereby, suppressing M2-like macrophage-dependent MCL pro-survival and proliferative effects. Additionally, GW2580, an orally bioavailable and selective CSF1R kinase inhibitor, reduced cell viability in both ibrutinib-sensitive and resistant primary MCL cells, indicating that targeting the CSF1/CSF1R axis could be an effective approach for treating ibrutinib-resistant patients. Inhibition of CSF1R with GW2580 significantly reduced monocyte survival and the M2-like marker CD163 on remaining viable monocytes [72].

### 3.2. Anti-Lymphangiogenic Effect of Lenalidomide in MCL

Abundant intratumor lymphatic vessels were observed in MCL specimens from patients and in an experimental mouse MCL xenograft lymphoma model [73]. A large number of VEGF-C^+^ macrophages in lymphoma samples from BCL patients and mice indicated that MCL-associated macrophages play a vital role in the induction of lymphomagenesis [74,75,76]. Meanwhile, lenalidomide reduce the number of macrophages in MCL. Additionally, monocyte depletion by clodronate liposomes significantly inhibited lymphomagenesis in MCL tumors and impaired MCL progression and dissemination in a mouse xenograft MCL model. Thus, lenalidomide may inhibit MCL growth by repressing recruitment of MCL-associated macrophages [73].

### 3.3. Trabectedin and Its Analogue Lurbinetedin

Trabectedin is a marine-derived compound that exhibit cytotoxic effects against macrophages in solid tumors. Lurbinectidine (PM01183) is an analog of trabectedin that has shown promising clinical activity in a broad range of clinical trials. For example, in PB, trabectedin or lurbinectedin significantly and selectively decreased the number of CD115^+^CD11b^+^ monocytes, especially the Ly6C^high^ subset that were predominantly recruited to peripheral inflammatory sites and tumor tissues [77]. In contrast, Ly6C^low^ monocytes, which correspond to human CD16^+^ monocytes, were absent from samples treated with trabectedin in CLL mouse studies [78]. Trabectedin primarily functions by decreasing the expression of mRNA encoding *Ccl2*, a monocyte chemoattractant, while increasing the expression of proinflammatory cytokines, such as *Ifna, Tnfa, Il12a*, and *Il6*, which facilitate the polarization of macrophages toward the classic M1 antitumor phenotype, evoking T-cell responses and interfering with B-cell activity [79]. The CCL2 (monocyte chemoattractant protein-1 [MCP-1])/CCR2 axis is implicated in the recruitment of myeloid cells to the tumor sites. As such, several antitumor strategies target this axis [80]. For example, trabectedin exhibited antileukemic effect in a xenotransplanted CLL murine model, and was related to a significant reduction in CD11b^+^F4/80^+^ TAMs in the spleen and peritoneal exudate, with selective depletion of CD11b^+^CD206^+^ M2-like protumor TAMs in the peritoneal exudate [78]. In fibrosarcoma tumor tissues, trabectedin or lurbinectedin reduced the number of macrophages and CD31^+^ blood vessels [81]. Meanwhile, in vitro, lurbinectedin reduced monocyte viability by inducing caspase-8-dependent apoptosis of human purified monocytes and trabectedin and lurbinetin significantly inhibited the secretion of inflammatory chemokines and growth factors, such as CCL2, CXCL8, and VEGF [81].

### 3.4. Artesunate

Artemisinin is a semi-synthetic compound of the sesquiterpene lactose drug family and is obtained from a Chinese plant used in traditional Chinese medicine and is known for its antimalaria properties [82]. Artesunate is a more stable and soluble derivative of artemisinin. In human monocytes obtained from PB mononuclear cells (PBMNCs), artesunate treatment induced an increase in inflammatory monocytes (CD14^high^CD16^−^) with high HLA-DR expression, NO production, and release of CCL2/MCP-1 and IL-1β. Thus, artesunate elicited an M1-like phenotype similar to that evoked by lipopolysaccharide treatment. Moreover, artesunate treatment reduced CD206 and CD163 expression and abolished the non-classical monocyte population (CD14^−^CD16^high^), while decreasing the intermediate subset (CD14^high^CD16^+^). This phenotypic change in monocytes was attributed to reduced phosphorylation of Janus kinase 2 (JAK2) and STAT3 [83].

## 4. Vaccination

### 4.1. Idiotype (Id) Vaccination for FL Patients

In 2002, Timmerman and Levy reported the results of long-term follow ups of 35 patients with FL treated with idiotype (Id)-pulsed DC vaccinations [84]. Ten patients with measurable relapsed or residual tumors after chemotherapy were enrolled in the pilot phase of the study followed by a cohort of 25 patients with advanced-stage FL treated in first remission after initial cytoreductive chemotherapy. Twenty-three patients completed a series of 3 monthly intravenous (IV) administrations of antigen-pulsed DCs followed by a fourth administered 2–6 months later and were evaluable for IRs. All 23 patients had humoral and cellular responses to the highly immunogenic carrier protein KLH. Of the 23 patients 15 (65%) induced T-cell anti-Id IRs or humoral anti-Id responses. Among the 12 patients evaluable for IRs, 4 induced cellular anti-Id responses, and 6 developed humoral anti-Id antibodies (4 raised high titers of IgG class and two IgM class). Following the administration of a booster vaccination of Id protein coupled to KLH, the patients who were resistant to, or had relapsed after the primary Id-pulsed DC vaccination, achieved complete tumor regression. 

Shuster et al. reported the results of a phase 3 trial, in which the primary endpoint was disease-free survival (DFS), to define the clinical benefit of vaccination with hybridoma-derived autologous tumor Ig conjugated to KLH administered with GM-CSF [85]. Of the 234 enrolled patients, patients achieving CR/CR unconfirmed (CRu) [86] following prednisone, doxorubicin, cyclophosphamide, and etoposide chemotherapy (*n* = 177, 81%) were randomly assigned to receive either the Id vaccine (Id-KLH with GM-CSF) or the control (KLH alone with GM-CSF). At a median follow up of 57 months, for 117 patients who received at least one blinded vaccination, the median DFS after randomization was 44 months for the Id-vaccine arm (*n* = 76) versus 31 months for the control arm (*n* = 41) (hazard ratio, 0.62; 95% confidence interval, 0.39–0.99; *p* = 0.047). In an un-predefined subgroup analysis, median DFS was significantly prolonged for patients receiving IgM-Id (53 months versus 29 months; *p* = 0.001), however, not for those receiving the IgG-Id vaccine (35 months vs. 32 months; *p* = 0.807) in comparison with isotype-matched control patients. This suggests that the isotype of the Fc region of Ig might affect the immunogenicity of Id vaccination. It is conceivable that the benefit of Id vaccination is discernible only in patients with minimal residual disease (MRD) following chemotherapy [85]. However, this study was initiated before rituximab had become standard of care for FL patients. Hence, considering that anti-CD20 mAbs blunt memory humoral IRs, evaluation of the efficacy of vaccinations administered simultaneously or following anti-CD20 MoAbs will be intricated, and thus future trials of vaccination for FL need to be well-designed.

Unfortunately, another phase 3 study, in which the primary endpoint was PFS, conducted by Levy et al. of a specific vaccine called MyVax, which was composed of Id chemically bound to KLH with GM-CSF, failed to show the clinical benefit of the specific vaccine [27]. That is, at a median follow-up of 58 months, no significant difference was observed between the groups in either PFS or time to subsequent anti-lymphoma therapy. However, patients who mounted higher anti-Id humoral response after the fourth immunization than that detected in the preimmunization serum plus two standard deviations from the mean of replicate wells at 4-fold greater dilution (IR-positive) had both superior PFS and time to subsequent anti-lymphoma therapy compared with IR-negative patients [27].

### 4.2. Vaccination for MCL Patients

By combining whole-exome sequencing of DNA and targeted Ig variable-region direct sequencing, and proteomic analysis of tumor-derived major MHC class I (MHC-I) and MHC-II ligands by immunoprecipitation and liquid chromatography/tandem mass spectrometry, Khodadoust and Alizadeh, et al. identified neoantigenic peptides exclusively derived from the lymphoma Ig heavy- or light-chain variable regions from 17 patients with untreated MCL [44]. However, no mutated peptides were recovered from non-Ig somatic mutated genes. Somatic mutations within the Ig variable region were mostly presented by MHC-II; 93% (105/113) of MHC-II bound peptides that spanned the junction of complementarity determining region 3 (CDR3) and framework region 3 (FR3). CD4^+^ T cells specific for Ig-derived neoantigens were isolated from PB and were found to mediate the killing of autologous lymphoma cells [44]. They also found 11 Ig-derived neoantigens presented by MHC-I from 4 out of 7 patients with FL or DLBCL. MHC-II presentation of Ig neoantigens was detected in all patients examined, with a total of 70 discovered class II neoantigens. The most frequently presented region of the Ig heavy chain was the FR3 [45]. They next tried to increase Ig neoantigen presentation by activating B-cell lymphomas with a TLR9 agonist to facilitate MHC II presentation [87]. In fact, the activation of MCL increased MHC-II expression (but not MHC-I), and global recovery of both MHC-I and MHC-II ligands was potentiated by TLR9 stimulation [44]. 

A phase1/2 trial of vaccination with irradiated, CpG-ODN PF-3512676-activated tumor cells was conducted for MCL patients who achieved remission after immunochemotherapy. The results demonstrated that vaccination with CpG-ODN-stimulated autologous tumor cells followed by adoptive transfer of vaccine-primed lymphocytes after ASCT was feasible [88]. The pre-specified primary endpoint was MRD, which was defined by Ig high-throughput sequencing [89] in the PB in a year after ASCT. Among the 45 evaluable patients, the 1-year freedom from MRD rate was 89%. With a median follow up of 4.6 years, the median time to progression (TTP) from initial chemotherapy was 8.1 years [88]. PBMNCs were co-cultured with CpG-ODN-activated autologous MCL cells and evaluated for tumor-specific IRs as measured by CD137 (also referred to as 4-1BB) expression on T cells, a sensitive marker for antigen engagement [90]. Vaccination evoked antitumor CD8^+^ T cell responses in 89% (40/45) of evaluable patients. After vaccination, 14 of 35 patients (40%) showed a CD8^+^ T cell response by expanding tumor-specific CD137^+^CD8^+^ T cells in PBMNCs. Forty percent (4/10) of patients administered circulating tumor cell-derived vaccines and 40% (10/25) of patients administered the LN-derived vaccines, produced memory CD8^+^ T cell responses. Patients producing such a memory CD8^+^ T cell response had a significantly longer TTP. Furthermore, the vaccine product stimulated with CpG significantly boosted the Ig neoantigen presentation and assumed that Ig neoantigens may be a cardinal antigen of the vaccine response [44]. A noticeable finding was that one patient with a high biologic Mantle Cell Lymphoma International Prognostic Index risk score [91] and *P53* mutation remained without detectable MRD over 43 months. Lymphoma Ig neoantigen-specific CD8^+^ and CD4^+^ T cells were identified in the PB over 2 years after vaccination completion/vaccine-primed lymphocyte reinfusion. After ex vivo expansion, these neoantigen-targeting T cells were capable of killing co-cultured lymphoma cells. Additionally, CpG induced the expression of programmed death ligand 1 (PD-L1) to a variable degree in tumor cells and PD-L1 expression in tumor cells after CpG activation was significantly correlated with TTP as well as overall survival. Intriguingly, patients with low PD-L1-expressing (PD-L1^low^) tumor vaccines generated a higher memory CD8^+^ T cell response (45%, 9/20) than patients with PD-L1^high^ tumor vaccine products (25%, 3/12) [88].

In summary, the 1-year freedom from MRD rate of the MCL patients vaccinated with irradiated, CpG-ODN activated tumor cells followed by adoptive transfer of vaccine-primed lymphocytes after ASCT, was 89%, while the median TTP from initial chemotherapy was 8.1 years.

## 5. Oligodeoxynucleotides Containing Unmethylated CpG Motifs (CpG-ODN) 

### 5.1. CpG-ODN

Efficient antigen cross-presentation by DCs is a prerequisite for the generation of an effective cytotoxic CD8^+^ T-cell-mediated anticancer response [92]. However, DCs at tumor sites are frequently dysfunctional and incapable of efficiently priming T cells [93]. CpG-ODN activates DCs [94,95,96,97] and B cells [98,99,100,101] by increasing the expression of costimulatory molecules such as CD80, CD86, and CD40, as well as MHC-II [102,103,104], and by triggering proinflammatory cytokine production [105]. TLR9 exists in rodents but not in primate macrophages or myeloid DCs [106]. In humans, TLR9 is expressed exclusively by plasmacytoid DCs (pDCs) and B cells [97,107,108,109,110]. 

CpG-ODN triggers an innate IR characterized by T-helper cells, type 1 (Th1) [111,112,113], and pro-inflammatory cytokine production [105,114,115,116,117]. Their utility as vaccine adjuvants was assessed in a plethora of clinical trials. Results imply that CpG-ODN ameliorates antigen presentation [118] and the generation of vaccine-specific cellular and humoral responses (Figure 3) [113]. Successful vaccination requires an intricate chain of immune interactions. Professional APCs, such as DCs and macrophages, as well as B cells take up the vaccine antigen. The antigen is, in turn, digested, and the fragments are presented to T cells. Antigen-activated CD4^+^ T cells promote antigen-specific B cells (Figure 3). Antigen-specific memory B and T cells yield, and sustain, long-term protection from subsequent antigen challenges [119,120,121,122].

DNA originating from bacteria can evoke the activation of B cells [98,123], NK cells [115,124], and macrophages [125]. Bacterial DNA contains more unmethylated CpG dinucleotides than vertebrate DNA because of CpG repression and methylation in 80% of the CpG in vertebrates [126]. Accordingly, it is plausible that lymphocyte activation by the CpG motif in bacterial DNA indicates an immune defense that discriminates bacterial DNA from host DNA [98]. Select synthetic CpG-ODN has immunogenic effects similar to those observed with bacterial DNA [126]. CpG-ODN stimulates monocytes, macrophages, and DCs. Conversely, vertebrate DNA does not activate lymphocyte.

TLR9 is expressed in the endosomal compartments of pDCs and B cells [127]. The immunostimulatory effects of CpG-ODN have direct and indirect effects on the immune system, such as B-cell proliferation and Ig production by B cells (Figure 2) [98], secretion of IFN-α/β (corresponding to a type I IFN) [105], IL-12 by macrophages [117] and pDCs [96], and tumor necrosis factor (TNF)-α by pDCs [105], and secondarily IFN-γ secretion from NK cells induced by IL-12 and IFN-α [128] (Figure 3). Then, these cytokines can elicit the differentiation of naïve CD4^+^ T cells into Th1 cells upon encountering specific antigens, leading to an adaptive immune response (Figure 3) [129,130,131]. The phosphodiester backbone of native DNA is quickly degraded by serum and cellular nucleases. Thus, in vivo applications with CpG-ODNs commonly use the nuclease-resistant, phosphorothioate-modified backbone (Figure 3, in box) [132], enhancing their cellular uptake and remarkably protracting the in vivo half-life of modified ODNs, although some non-sequence-specific effects are induced [129].

Thus, the immune effects of CpG-ODNs may be divided into the following two stages: an early stage of innate immune actuation and a later stage of adaptive IR potentiation (Figure 3). Within minutes of exposure of B cells or pDCs to CpG-ODNs, the ODNs enter an endosomal compartment in which they engage TLR9 (Figure 3), resulting in the activation of cell signaling pathways that culminate in the expression of costimulatory molecules and the upregulation of C-C chemokine receptor type 7 (CCR7). CCR7 leads to cell trafficking to the T-cell zone of the LNs, and production of Th1 accelerating the expression of various chemokines and cytokines, such as *macrophage inflammatory protein-1α* (also known as *CCL3*), and other IFN-inducible genes [129]. pDCs secrete type I IFN and mature into highly effective APCs [105]. These CpG-induced type I IFNs, cytokines, and chemokines trigger, within hours, a wide range of secondary effects, including NK-cell activation (Figure 3), and enhanced expression of FcRs, such as polymorphonuclear leukocytes, leading to an increase in antibody-dependent cell-mediated cytotoxicity (ADCC).

Collectively, this innate immune activation and pDC maturation is followed by the generation of adaptive IR. There are two important phases of the IRs: the antigen-priming phase and the effector phase, the latter of which antitumor T cells migrate to and eliminate the tumor. This T-cell stimulation does not depend on the presence of the typical DNA motif but is relevant to the presence of guanosine in the sequence.

To activate T cells, APCs must present two concomitant signals: the first signal through cognate antigen (signal 1), and the second signal by costimulatory molecules (signal 2) (Figure 2). CpG-ODN–loaded tumor cells potentiated the phagocytic ability of APCs and induced the same APCs to express higher levels of costimulatory molecules. More F4/80^+^ macrophages and DCs from C57BL/6 mice phagocytosed CpG/H11 tumors than H11 alone [133].

Link and Weiner, et al. reported the results of a phase 1 trial of a class B (also referred to as Type K) CpG-ODN (24 nucleotides in length) PF-3512676, also known as CPG 7909, in patients with previously treated lymphoma [134]. PF-3512676 was administered at doses in the range of 0.01–0.64 mg/kg once a week as a 2-h IV infusion for 3 consecutive weeks. Class B CpG-ODNs directly activate B cells and pDCs; however, induce much less IFN-α secretion attributable to the secondary activation of NK cells compared with class A CpG-ODNs (Figure 3, in box) [135]. Twenty-three patients were enrolled, and the median number of previous treatments was four. Infusion-related toxicities included G1 nausea, hypotension, and IV catheter discomfort, and G3/4 hematologic AEs documented more than once were anemia, thrombocytopenia, and neutropenia (Table 1). Additionally, six patients experienced transient lymphopenia. Although no clinical responses were observed on day 42, among 21 patients with available serum samples, 5 patients showed elevated levels of IgG with a > 5% but ≤ 20% change from baseline, and a slight increase in serum monoclonal IgM levels was observed in 2 of the 3 patients with preexisting IgM paraproteins [134].

### 5.2. CpG-ODN Combined with an Anti-CD20 Antibody Rituximab

In an immunocompetent murine model, treatment with CpG-ODN combined with an antitumor MoAb is effective in preventing lymphoma growth. CpG-ODN alone did not have any effect on the survival of mice bearing 38C13 cell tumors (a murine B-cell lymphoma cell line). This antitumor effect was less pronounced when the MoAb treatment was combined with an identical ODN containing methylated CpG dinucleotides [136]. Leonard, Link, and Weiner, et al. reported the results of a phase 1 trial of IV or subcutaneous (SC) administration of PF-3512676 (CPG 7909) within 24 h following four weekly IV infusions of rituximab in R/R CD20^+^B-cell NHL [137]. PF-3512676 was administered weekly for 4 weeks via 2-h IV infusion at 4 dose levels: 0.04, 0.16, 0.32, or 0.48 mg/kg or via SC injection at 0.01, 0.04, 0.08, or 0.16 mg/kg doses. An additional extended-treatment cohort received 4 weeks of 0.24 mg/kg SC PF-3512676 in combination with rituximab followed by SC PF-3512676 alone weekly for 20 weeks. Fifty patients (including six MCL, 22 FL, 14 DLBCL, and eight other indolent B-cell lymphomas in IV or SC cohorts), who received a median of three prior therapies, were enrolled. Treatment-related AEs occurred in 58% (11/19 patients) of the IV cohort, opposed to 79% (15/19) in the SC cohort. Systemic flu-like syndrome and injection-site reactions (SC cohort only) were the most common AEs, although they were mild or moderate (Table 1). Overall, 24% (12/50) of patients achieved OR, and 50% (6/20) were in the extended-treatment cohort [137].

Friedberg et al. reported the results of a phase 1 study evaluating 4 dose levels of 1018 ISS with rituximab in 20 patients with R/R FL. Patients received SC CpG-ODN once a week for 4 weeks, starting after the second of four weekly rituximab infusions. The expressions of several IFN-α/β-inducible genes were increased 24 h after 1018 ISS injection in a dose-dependent manner in higher-dose (0.05, 0.2, and 0.5 mg/kg) groups (Table 1) [138]. In a phase 2 trial, 45% of patients showed an elevation in T-cell and DC counts in skin biopsies of 1018 ISS injection sites 24 h after CpG-ODN injection. Of the 23 enrolled R/R FL patients, 48% achieved a PR, CR, or CRu [139]. Six additional patients (26%) maintained stable disease on day 90. The percentage of patients alive without progression at 1 year from study entry was estimated to be 41%. Enzyme-linked immunosorbent spot (ELISPOT) assay showed that 25% of patients had more than twofold elevation in IFN-γ secretion from T cells in PBMNCs after stimulation on day 29 in comparison with the baseline. In addition, 35% of patients had more than a 10% increase in ADCC as measured by a chromium release assay on day 29 as compared with the baseline. Among three examined interferon-α/β (namely, a type I IFN)-inducible genes, namely, *interferon induced protein with tetratricopeptide repeats 2 (IFIT2)*, *CCL2*, (formerly known as *monocyte chemoattractant protein 1 (MCP1)*), and *CXCL10*, (formerly known as *IFN-gamma-inducible protein of 10 kilodalton (IP-10)*) that were measured by quantitative polymerase chain reaction, patients who had more than a twofold increase in *CXCL10*, which codes a chemoattractant for CD3^+^ T cells, expression following 1018 ISS therapy were more likely to respond to the therapy on day 90 compared with patients who had less than a twofold changes in expression (odds ratio = 14, *p* = 0.03). Pre- and post-therapy paired biopsies of tumor tissues, including LNs and bone marrow (BM), were performed in six patients, all of which exhibited a significant number of infiltrated CD3^+^ (CD8^+^ > CD4^+^) T cells and macrophages following the combination treatment (Table 1) [139].

### 5.3. CpG-ODN Combined with Anti-OX40/CD134 and Anti-CTLA4/CD152 MoAbs

Houot and Levy combined IT injection of CpG-ODN with MoAbs against T-cell targets, especially anti-OX40/CD134 and anti-CTLA4 MoAbs to augment T-cell modulation [140]. After completion of treatment with daily IT CpG injection for 5 days with intraperitoneal injections (IP) of anti-OX40/CD134 and anti-CTLA4/CD152 antibodies on days 1 and 5, total splenocytes were co-cultured in the presence of irradiated A20 tumor cells plus ant-CD28 MoAb overnight. From these cultures, antitumor IFN-γ–producing T cells were generated 5 days after SC inoculation with A20 tumor in Balb/c mice, and these IFN-γ–producing T cells primarily comprised the CD44^high^ memory T-cell, leading to immune protection from tumor re-challenge over 100 days after regression of the primary tumors. Additionally, this combination decreased the percentage of regulatory T cells (Tregs) (Forkhead boxprotein P3 (FoxP3)-expressing CD4^+^ T). However, depletion of either CD4^+^ or CD8^+^ T cells completely ablated the effect of CpG-ODN combined with anti-OX40/CD134 and anti-CTLA4/CD152 antibodies [140]. Thus, CD4^+^ and CD8^+^ T cells are required for triplet combination therapy.

### 5.4. CpG-ODN Combined with BTK Inhibitor Ibrutinib

Ibrutinib also inhibits other members of the Tec family of tyrosine kinase, such as IL-2-inducible T-cell kinase, an important member of the signaling pathway in T cells, especially the T-helper cell, type 2 (Th2) subset of CD4^+^ T cells [6]. Therefore, ibrutinib shifts the balance of T-cell responses toward the more therapeutically effective Th1 subset. To investigate whether ibrutinib can enhance the effect of IT CpG-ODN injection, Sagiv-Barfi and Levy, et al. aimed to eradicate mouse lymphoma at the injected sites as well as at distant sites [141]. To this end, they administered ibrutinib daily at a 6 mg/kg dose for eight days beginning eight days after tumor implantation. The results suggested that ibrutinib induced immunogenic cell death (ICD) of lymphoma cells, although no definitive evidence was presented in the study. Moreover, simultaneous stimulation of APCs occurred in the tumor microenvironment. They postulated that ibrutinib would kill tumor cells and liberate tumor antigens to be taken up by the CpG-ODN-potentiated APCs, which in turn would evoke a more robust T-cell IR [141]. IT injection of CpG-ODN alone regressed the tumor at the injected site without any effect on the distant tumor. However, concomitant administration of IT CpG-ODN with systemic ibrutinib followed by ibrutinib alone for a further three days led to complete and sustained regression of tumors. Furthermore, surviving mice were resistant to re-challenge with H11 tumor cells on day 120. A similar effect was observed with ibrutinib-sensitive BL3750 but not with ibrutinib-insensitive A20 tumors. These results were compatible with the assumption that direct tumor killing by ibrutinib, followed by a systemic IR triggered by antigen presentation at the CpG-ODN injection site, resulted in a systemic antitumor response. Systemic regression of the tumor mediated by the combination of IT CpG-ODN and systemic ibrutinib occurring as soon as the initiation of the treatment was reminiscent of the involvement of the innate immune system rather than the adaptive immune system. In severe combined immunodeficiency (SCID) and nude mice, which lack T/B cells but are replete in myeloid-derived/NK cells and are devoid of functional T cells but are replete in B cells, respectively, the same combination treatment failed to cause tumor regression either at the treated or non-treated site. Thus, the former experiment implies that T cells and/or B cells were required for the effect of IT CpG-ODN combined with systemic ibrutinib, while the latter suggests that T cells rather than B cells are required for the effect of the combination [141]. Finally, both CD8^+^ and CD4^+^ T cells were shown to be required for the therapeutic effect because depletion of CD4^+^, CD8^+^, or both T cells by specific antibodies abolished the antitumor response in immune-competent mice. In addition, on day 4 of post-treatment, approximately 6% of CD4^+^ T cells and 11% of CD8^+^ T cells of total lymphocytes produced IFN-γ in response to co-culture with H11 tumor cells. Furthermore, tumor-specific, IFN-γ-producing CD8^+^ and CD4^+^ T cell responses were detected in the memory T-cell subset, defined by CD44^high^ obtained from the mouse spleen. After exposure to ibrutinib and CpG-ODN, splenocytes or purified T cells co-cultured with H11 cells induced intracellular IFN-γ, and higher frequencies of the cells responded in the presence of splenic APCs than without them [141]. 

### 5.5. CpG-ODN Combined with Low-Dose Radiation

Brody and Levy, et al. reported the results of a phase 1/2 trial of PF-3512676, which evoked systemic anti-lymphoma clinical responses [32]. Fifteen patients with relapsed low-grade B-cell lymphoma, including 13 FL and two marginal zone lymphomas (MZL), were enrolled in the study and treated with an IT injection of 6 mg PF-3512676 immediately before the first radiation in the same tumor site as irradiation. The radiation dose was 4 Gy over 2 consecutive days, known to kill some tumor cells sparing the APCs in the tumor microenvironment, to a solitary tumor. The IT administration of CpG-ODN was well-tolerated; the only observed AEs were G2 injection site reactions and G1/2 flu-like reactions, which lasted 24–72 h after CpG-ODN injection (Table 1). One FL patient achieved a CR, three others (two FL and one MZL) obtained PRs, and an additional two FL patients had stable disease, whose disease continued to regress for periods significantly longer than those achieved with prior therapies. Meanwhile, a previous study investigating the effect of radiotherapy alone on R/R FL patients, reported that CR was achieved in 73% (11/15), and PR in 13% (2/15) with 30 Gy (range: 2–40 Gy) median dose of radiation [142]; however, the radiation doses implemented in these studies differed. CD137 (also referred to as 4-1BB) is the most sensitive marker, especially in memory (determined by CD45RO^+^) CD8^+^ T cells, among the measured activation markers CD137/4-1BB, IFN-γ, IL-2, and TNF. FL are tumors rich in Tregs (especially follicular regulatory T cells) due to FL B-cell secretion of chemokine attractants for Tregs and follicular helper T cells (Tfhs) [143]. Pre-vaccination PB lymphocytes (PBLs) were cultured either with media or with autologous, irradiated, CpG-ODN-activated lymphoma cells, and the expression of CD4, CD25, and FOXP3 was assessed. Although the baseline proportion of Tregs in the tumor of each patient and PB seemed to be unrelated to clinical outcomes, PFS in patients who indicated a 7-fold (mean, 6.6; range, 4.5–12.8) increase in Treg (*n* = 5) (Treg inducers) was significantly shorter than that in non-Treg inducers (mean, 1.5; range, 0.9–2.1) (*n* = 9). One patient, who achieved a PR lasting 16 months and was re-treated with a higher dose of PF-3512676 after recurrence, achieved a second PR. The magnitude of the second response was slightly higher and more rapid than that obtained after the initial vaccination with CpG-ODN. Local low-dose irradiation seemed to induce ICD and systemic immune response, i.e., the so-called “abscopal effect [144,145,146,147,148,149,150,151].” The relationship between clinical responses and fewer prior therapies suggested that vaccination with a CpG-ODN might be more effective in previously untreated patients [33]. Whether Treg-inducing tumor pre-in situ vaccination would be a powerful clinical indicator for appropriate candidates for in situ vaccination should be further explored in larger studies in the future.

Furthermore, Frank and Levy et al. conducted a phase 1/2 trial of IT SD-101, a class C CpG-ODN (Figure 2, in box), which elicits high levels of IFN-α and DC maturation, and low-dose radiation in patients with untreated indolent lymphoma [152]. Twenty-nine patients were enrolled in the trial, three (10%) patients with Stage II disease, 6 (21%) stage III, and 20 (69%) stage IV. All patients received 2 Gy per day on days 1–2 of radiation followed by 5 weekly IT injections of SD101 that induced high levels of IFNα and DC maturation, at a single tumor site. The primary endpoints were safety and induction of IFN-regulated gene expression in PB cells. The secondary endpoints were clinical efficacy and modulation of the immune microenvironment at the treated as well as untreated sites of the tumor. All 29 enrolled patients were evaluable for clinical response with a median follow up of 12 months, and 26 of the 29 treated patients showed a reduction in overall tumor burden, with 7 patients achieving PR and 1 CR determined by the sum of the products of the diameters in all target lesions [153]. Fine-needle aspiration biopsies were performed on day 9 from the treated site, revealing a significant reduction in malignant B cells and an increase in CD3^+^, CD8^+^, and CD4^+^ T cells. Moreover, CD4^+^ T-cell subset analysis showed a significant decrease in Tfhs and Tregs, as well as an increase in effector CD4^+^ T cells. The distal, non-injected site was also biopsied, if available, and intriguingly, showed a low initial proportion of proliferating (defined by Ki67^+^) CD8^+^ T cells and granzyme B^+^ CD8^+^ T cells were associated with better clinical outcomes. It was discovered that a low baseline proportion of CD4^+^ Tregs and post-treatment MHC-II expression on the tumor cells at the treated sites were significantly related to better clinical responses at distal sites. Collectively, the results corroborated the previous finding that systemic tumor elimination by TLR9 agonists was partially CD4^+^ T cell-dependent [154]. *Cyclic-adenosine monophosphate response element binding protein* (*CREBBP)*, which is associated with the downregulation of MHC-II [155], is known as the second most frequently mutated gene in FL [156,157], and therefore, it is very interesting to investigate whether CpG-ODN in addition to low-dose irradiation, which induces abscopal effect, will have the advantage of showing a systemic anti-lymphoma effect even in FL patients with *CREBPP* mutations.

### 5.6. CpG-ODN Combined with Radioimmunotherapy

Witzig and Weiner et al. conducted a phase 1 trial that investigated four dose levels of IV CpG 7909 combined with a standard dose of ^90^Y-ibritumomab tiuxetan radioimmunotherapy (RIT). The patients received two doses of CpG 7909 before RIT and the remaining two doses after RIT. The two doses of CpG 7909 following RIT were designed to stimulate immunity after the RIT had induced lymphoma cell death and antigen release. The examined doses of CpG 7909 were 0.08, 0.16, 0.32 (six patients each) and 0.48 mg/kg (12 patients). The ORR was 93% (28/30), which was higher than those in previous RIT studies wherein ORRs were generally within the 80% range. Moreover, the median TTP in patients treated with a CpG 7909/RIT combination was 40.5 months, which was longer than 12–18 months TTP reported for relapsed patients enrolled in most single-agent RIT studies. Additionally, CR was achieved in 63% (19/30) of the patients with a median PFS of 43 months. Serum cytokine measurements revealed a significant reduction in IL-10 and TNF-α and elevation in IL-1β. Dosimetry further revealed that CpG 7909 did not have any impact on tumor or normal organ distribution of the radionucleotide [158].

Currently, a phase 1b/2 study of IT SD-101, a class C CpG-ODN in combination with ibrutinib and local radiation is underway for patients with R/R Grade1-3A FL, MCL or MZL (NCT02927964). Additionally, a phase 1 study of IT SD-101 in combination with anti-OX40 antibody BMS 986178 and local low-dose radiation therapy using high-energy X-rays, is ongoing for patients with low-grade B-cell NHL (NCT03410901).

**Table 1 cancers-14-00141-t001:** Clinical trials and mouse studies of CpG-ODN alone or in combination with other modalities.

Combination	ODN Class	Name of CpG (Route)	Phase	Object	N	Efficacy (%)	IR (%)	AEs	Authors [Ref]
CpG-ODN alone	B(=Type K)	PF-3512676 [=CPG7909] (IV)	P1	R/R NHL	23	No clinical responses	IgG elevation in 5 pts	Anemia Thrombocytopenia Neutropenia Transient lymphopenia Dyspnea Chills/rigors Nausea/vomiting Fever Hypotension Fatigue Pain Intravenous catheter discomfort ALT increased Hyperglycemia Proteinuria	Link, and Weiner et al. [134]
CpG + R	B(=Type K)	PF-3512676 [=CPG7909] (IV or SC)	P1	Relapsed CD20^+^ B-NHL	50	12OR (24%)	NA	Systemic flu-like syndromes ^#1–1^, Fatigue or rigors Pyrexia Malaise Injection-site reactions ^#2–1^ Neutropenia Pain in limbs Generalized pruritus Peripheral neuropathy Sjögren’s-like syndrome ^#3^	Leonard, Link, and Weiner et al. [137]
CpG + R	B(=Type K)	1018 ISS (SC)	P1	R/R FL	20	6OR [1CRu+5PR] (32%)+ 13SD Median PFS in responding pts: 12 mos	Induction of IFN-α/β—inducible genes	Allergic reaction ^#4^ Headache Fatigue Atypical pneumonia	Friedberg, et al. [138]
CpG + R	B(=Type K)	1018 ISS (SC)	P2	R/R FL	23	11OR [CR/CRu or PR] (48%) + 6SD (26%) Median PFS: 8.8 mos	T-cell and MØ infiltration in injection sites IFN-γ—secretion increased from T cells in PB (25%) ADCC increased (35%) Increased expression of IFN-α/β-inducible genes (*IFIT2, CXCL10, CCL2*) (≥60%) Increased *CXCL10* expression in PB in responders	Fatigue Rituximab infusion reaction Erythema at the injection sites	Friedberg, et al. [139]
CpG + αOX40 + αCTLA4	B(=Type K)	CpG1826 (IT)	mice	A20 tumor cells	NA	Immune protection from tumor re-challenge over 100 days	IFN-γ-producing, memory T cell increased Treg decreased	NA	Houot and Levy et al. [140]
CpG + ibrutinib	B(=Type K)	CpG1826 (IT)	mice	H11 and B3750 tumor cells	NA	Tumor regression in at the distant site	% increase in tumor-specific IFN-γ-producing, memory T cells	NA	Sagiv-Barfi and Levy et al. [141]
CpG + LD RT	B(=Type K)	PF-3512676 (IT)	P1/2	Relapsed low-grade B-NHL	15	1CR, 2PRs	Induced CD137/4-1BB expression and increased intracellular IFN-γ, IL -2, and TNF in memory CD8^+^ T cells in PB Treg induction in PB and tumor	Injection site reaction ^#2–2^ Flu-like reactions Arthralgia Myalgia	Brody and Levy et al. [32]
CpG + LD RT (2 Gy x 2)	C	SD-101 (IT)	P1/2	Untreated iNHL	29	26 reductions, including 1CR, 7PRs	Increased T cell and decreased Tfh and Treg in injected tumors	Flu-like systemic reactions ^#1–2^ Myalgia Fever Nausea/vomiting Diarrhea Injection-site reaction ^#2–3^ Neutropenia Confusion Decreased appetite Night sweating Thrombocytopenia	Frank and Levy et al. [152]
CpG + RIT	B(=Type K)	PF-3512676 [=CPG7909] (IV)	P1	R/R CD20^+^B-NHL	30	28OR (93%) 19CRs (63%) Median PFS: 43 mos DOR: 35 mos Median TTP: 40.5 mos	Decrease in IL-10 and TNF-α and increase in IL-1β in serum	Reversible myelosuppression (due to RIT) Pain on bulk neck lymph node (after infusion of CpG 7909)	Witzig and Weiner et al. [158]

^#1–1^ Were comprised of fever, fatigue, and headache. ^#1–2^ Consisted of malaise, chills, headache, fatigue, and fever. ^#2–1^ Included erythema, pain, edema, pruritus, bruising, and indurations. ^#2–2^ Included erythema, induration, and tenderness. ^#2–3^ Included erythema, swelling, and pain. ^#3^ Was composed of conjunctivitis and oral mucositis. ^#4^ Included hives, rhinitis, rigors, chills, and dyspnea. Abbreviations: ODN, oligodeoxynucleotide; IR, immune response; AEs, adverse events; IV, intravenous; P1, phase 1 study; R/R, relapsed or refractory; NHL, non-Hodgkin lymphoma; IgG, immunologlobulin G; pts, patients; aPTT, activated partial thromboplastin time; ALT, alanine aminotransferase; R, rituximab; SC, subcutaneous; B-NHL, B-cell NHL; OR, overall response; NA, not available; FL, follicular lymphoma; CRu, unconfirmed complete response; PR, partial response; SD, stable disease; PFS, progression-free survival; mos, months; IFN, interferon; CR, complete response; MØ, macrophage; PB, peripheral blood; ADCC, antibody-dependent cell-mediated cytotoxicity; IFIT2, interferon-induced protein with tetratricopeptide repeats 2; CXCL10, C-X-C motif chemokine ligand 10; CCL2, C-C motif chemokine ligand 2; αOX40, anti-OX40 antibody; αCTLA4, anti-cytotoxic T-lymphocyte-associated protein 4 antibody; IT, intratumoral; Treg, regulatory T cells; LD, low-dose; RT, radiotherapy; IL, interleukin; TNF, tumor necrosis factor; iNHL, indolent non-Hodgkin lymphoma; Tfh, follicular helper T cells; auto DCs, autologous dendritic cells; RIT, radioimmunotherapy; DOR, duration of response; TTP; time to progression.

## 6. CpG-ODN Containing Virus-like Particle (VLP) and CpG ODN Combined with Anti-PD-1 Ab

CpG-ODN is intrinsically vulnerable to nuclease and thus soluble TLR9 agonists delivered in situ may be rapidly degraded and diffuse out of the tumor [159]. Recently, Weiner, Krieg, and colleagues developed CMP-001, formerly known as CYT003 or QβG10, a virus-like particle (VLP), which consists of the recombinant Qβ bacteriophage capsid protein encapsulating synthetic G10, a class A CpG-ODN. VLPs are noninfectious, self-assembling, highly immunogenic delivery systems [160]. In vitro, CMP-001 produced proinflammatory cytokines including IFN-α only in the presence of Qβ-immune serum or recombinant anti-Qβ antibody. These results suggest that anti-Qβ opsonization of CMP-001 VLPs needs to increase the secretion of proinflammatory cytokines/chemokines from immune cells in vitro. This effect is predominantly mediated by pDCs in human PBMNCs. In vivo, IT CMP-001 extended the survival of Balb/c mice bearing A20 lymphoma, which constitutively expresses PD-L1 [161], and regression of both injected and non-injected tumors was dependent on whether mice could produce anti-Qβ antibody and whether CD4^+^ or CD8^+^ T cells existed [160]. Studies utilizing TLR9 agonists as immune adjuvants in cancer vaccines comprising of tumor-associated antigens showed strong clinical induction of antitumor CD4^+^ and CD8^+^ T cells. Moreover, a previous report indicated that antigen-specific CD8^+^ T cells induced by CpG-ODN-based vaccines express high levels of PD-1 [162]. They showed that the IT delivery of CPG-ODN containing the novel VLP combined with IP anti-PD-1 MoAb consequently augmented antitumor responses in injected sites as well as contralateral non-injected sites. IT CMP-101 alone enhanced T-cell infiltration in tumor-associated draining LNs only at injected sites, however, combination therapy increased the number of infiltrated T cells in injected and non-injected tumor-associated draining LNs [160].

Currently, a phase 1/2 study of CMP-001 combined with pembrolizumab is ongoing for patients with R/R lymphoma.

## 7. Vaccination with Immunogenic Cell Death Tumors

### 7.1. Vaccination with Autologous Lymphoma Cell-Loaded DCs

Di Nicola and Gianni reported the results of a pilot study of active immunization with DCs loaded with heat-shocked, irradiated autologous tumor cells and presented clinical and immunologic efficacy in some previously treated indolent NHL patients with measurable disease [163]. In patients who achieved PR and, thus, had T cells at residual tumor sites, vaccination significantly augmented the IFN-γ-producing T cell response to autologous tumor challenges. However, similar evidence was not observed in patients with stable disease or progressive disease. In one of the patients who achieved a CR after vaccination with HLA-A^*^02:01, ELISPOT assay showed an increased frequency of T cells secreting IFN-γ and IL-4 in PB in response to tumor-specific CDR1-, FR1-, and CDR3-derived peptides of a tumor-specific Ig heavy-chain sequences that were identified after the first vaccination. In addition, immunohistochemical examination of tumor biopsies using biotin-conjugated autologous serum samples disclosed a tumor-restricted humoral response only in the post-vaccination serum from one patient in CR and one in PR. Taken together, these results showed that vaccination with heat-shocked necroptotic (immunogenic regulated cell death [164,165,166,167]) tumor cells promoted the efficient transfer of tumor-associated antigens to the MHC-I processing pathway of APCs, induced T and B cell responses, and produced clinical benefits in indolent NHL patients with measurable diseases [163].

ICD [168,169,170] is preceded by relocation of the endoplasmic reticulum-resident chaperone calreticulin (CRT) to the plasma membrane, followed by surface expression of heat shock protein (HSP) 70 and HSP90, which serve as vehicles for peptide antigens. Using histologically different NHL cell lines, such as DoHH-2, SU-DHL-4 and SU-DHL-6, Zappasodi and Di Nicola et al. demonstrated that heat shock, γ-ray, or ultraviolet C ray, or their combined use increased surface translocation of CRT and cellular release of the chromatin-binding protein high-mobility group box 1 (HMGB1) and adenosine triphosphate compared with doxorubicin, a well-known chemotherapeutic agent that causes ICD and is included in a combination chemotherapy regimens, such as CHOP, and is frequently used as treatment for NHL patients. Furthermore, they conducted a study of DC vaccines pulsed with autologous lymphoma cells using all the treatment mentioned above for indolent NHL patients. Interestingly, the percentage of CRT-expressing cells and their intensity, defined by the mean fluorescence intensity of flow cytometry analysis, was significantly higher in dying lymphoma cells from responders. Apoptotic and necrotic bodies from tumor cells of responders expressed surface HSP90 at a significantly higher frequency and intensity in flow cytometry analysis compared with those from non-responders, although no significant difference was observed in the case of HSP70. Both HLA-I and HLA-II expression in dying lymphoma or HMGB1 released from these cells differed between responders and non-responders. Furthermore, after vaccination, the responders raised a higher number of antibodies specific to CRT and HSP90 than non-responders when the surface expression of CRT or HSP90 in dying lymphoma cells used antigen cargo for a DC-based vaccination. Consequently, the extent of CRT and HSP90 surface expression in DC antigenic cargo was significantly correlated with the clinical and immunological responses achieved [171].

### 7.2. IFN-α—Conditioned DC (IFN-DC) Vaccination

Montico and Dal Col et al. developed monocyte-derived DCs generated in the presence of IFN-α and GM-CSF (IFN-α–conditioned DC) with highly immunogenic tumor cell lysates (TCLs) obtained from lymphoma cells undergoing ICD. MCL and DLBCL cells were treated with 9-*cis*-retinoic acid (RA) and IFN-α-conditioned DC (RA/IFN-α) induced translocation of CRT to the cell surface along with CD47 downregulation, enhanced cell surface expression of HSP70 and HSP90, and increased HMGB1 release [172]. Cytotoxic T cells produced with autologous DCs pulsed with RA/IFN-α–TCLs more efficiently cognized and specifically lysed MCL or DLBCL cells. In particular, Balb/c mice vaccinated with A20 cells treated with RA/IFN-α were protected against re-challenge with A20 cells. Functional annotation analysis of activated gene signatures by RNA-sequencing of DCs loaded with RA/IFNα-TCLs involved “DC Maturation,” “IL-6 Signaling,” and “TLR Signaling,” related to maturation/activation of DCs, as well as “HMGB1 Signaling,” indicating ICD. Moreover, non-obese diabetic (NOD)/SCID mice reconstituted with human PBLs were vaccinated with autologous RA/IFN-α-TCL loaded IFN-α-conditioned DCs (IFN-DCs). ELISPOT assay of human cells collected from mouse spleens 7 days after the last boost immunization 14 days after the initial immunization in addition to boosting on day 7 exhibited higher numbers of CD3^+^ T cells secreting IFN-γ and Granzyme B, and tumor growth was significantly inhibited [172]. 

A previous study by Lapenta and Santini et al. revealed that an anti-lymphoma response against autologous lymphoma cells in a small group of FL patients induced by IFN-DCs was characterized by the reinvigoration of T cell responses with increased levels of IFN-γ and TNF-α production in vitro, preceded by NK cell activation by increased expression of cytotoxic receptors, such as NKp30, NKp44, and NKp46 and NK cell expansion, especially more brisk proliferation of CD56^brtight^ cells than CD56^dim^ cells, and by the early activation marker CD69 expression and exclusively large amount of IFN-γ intracellular production by NK cells [173]. The percentage of the degranulating cells (CD107a^+^) was remarkably increased in NK (CD56^+^CD3^−^) cells and CD8^+^ T cells after PBLs from FL patients were co-cultured for 14 days with autologous apoptotic lymphoma cell-loaded IFN-DCs, exhibited perforin-containing granules at the cell-cell contact site of CD56^+^ and CD8^+^ lymphocytes. The results implicate the potentiation of both innate and specific immune responses to autologous lymphoma cells. Furthermore, a polarization of IL-15 intracellular distribution was observed at the immunological synapses of DCs and NK cells. Moreover, MHC-I-related chain A and B (MICA/B) and membrane-bound IL-15 play a pivotal role in IFN-DC–mediated NK cell activation and early IFN-γ production because anti-MICA/B or anti–IL-15 neutralizing antibody remarkably diminished the expression of CD69 on NK cells [173].

### 7.3. Immunomodulatory Drugs Combined with Interferon-α-DC-Based Vaccination

Combined with IFN-DC-based vaccination, the immunomodulatory drug, lenalidomide, which was approved for R/R FL patients [2] and active for untreated FL [174] and MCL patients [28,175], increased the frequency of IFN-γ production by NK cells in response to Karpas-422, a DLBCL cell line harboring t(14,18) translocation [176]. Lenalidomide reduced IL-10 released from T cells and significantly reduced Treg frequency. Furthermore, PBL cultured for 14 days with the Karpas-422 cell-loaded IFN-DCs in the presence of lenalidomide displayed an increased frequency of immune synapse formation identified by actin-staining between CD8^+^ T cells and lymphoma cells. Finally, lenalidomide was administered to NOD/Scid mice bearing Karpas-422 cells, reconstituted with human PBL on day 0 and vaccinated with IFN-DCs loaded with apoptotic Karpas-422 on days 4, 11, and 18. Tumor growth was suppressed without apparent regression after either treatment alone. However, the combination of lenalidomide and vaccination significantly reduced tumor size and delayed tumor regrowth upon treatment discontinuation. Necroscopic examination revealed that, in control mice, lymphoma cells spread from the injection site to the axillary LNs, but that lymphoma cell dissemination to axillary LNs was diminished in the mice receiving treatment. In tumor tissues examined 14 days after the last immunization, no FOXP3^+^ T cells were identified in the mice treated with the combination therapy, whereas scarce FOXP3^+^ T cells were found in the corresponding tumors from mice treated with the IFN-DC vaccine alone. Consequently, the median number of tumor-infiltrating CD8^+^ T cells in the combination therapy group was far higher than that in the group receiving vaccination alone [176]. Hence, the design of a triple combination therapy comprising vaccination, TLR-agonists, and CAR T-cell therapy followed by maintenance therapy of vaccination and/or lenalidomide, may prove highly effective. 

## 8. STING Agonists (STINGa) 

Cyclic dinucleotides (CDNs) are another class of immune stimulators and induce APCs to produce cytokines and chemokines including type 1 IFN [177]. They activate the immune system via STING. STING is a receptor that perceives CDN generated by the microorganism or endogenously generated on cytosolic DNA detection by cyclic guanosine monophosphate and adenosine monophosphate synthase (cGAS) [178]. The cGAS-STING pathway is also implicated in the spontaneous immune recognition of tumors. In the case of injected IT, CDNs are capable of evoking a tumor-specific T cell response. They provoke APCs to generate cytokines and chemokines, such as a type 1 IFN. IT injection of CDNs has been applied in several solid tumors such as melanoma, colorectal cancer, pancreatic cancer, and breast cancer [177,179,180,181]. The local effect of STINGa was ameliorated by adding agents that activate APCs including CpG-ODN [180], glucocorticoid-induced TNF receptor-related protein (GITR) is an immunomodulatory agent that synergizes with a STINGa as a therapeutic in situ vaccination of lymphoma [49]. Following the STINGa synthetic dithio-modified cyclic diadenosine ADU-S100 and anti-GITR treatment, PD-1 is expressed on a high number of CD8^+^ T cells. The percentage of CD4^+^ FoxP3^+^ T cells was lower in the tumors treated with STINGa, implying that tumor-infiltrating Tregs might lose FoxP3 expression following STINGa treatment. This combination increased the proportion of CD8^+^ T cells among the CD3^+^ T cells in the draining LN of the treated tumor, leading to an increase in the CD8/Treg ratio. Moreover, the combination increased the percentage of CD69-, Ki67-, or inducible T-cell co-stimulator (ICOS)-expressing cells as well as activated B cells. In the distant, non-injected tumor, the combination increased CD8^+^ T-cell activation (defined as CD69^+^, Ki67^+^, or ICOS^+^) as early as one day after treatment, and T-cell infiltration was increased one week after treatment. The percentage of PD-1^low^ cells was higher one week after treatment. Accordingly, the triple combination: IT injection of a STINGa and IP injection of an anti-GITR antibody and anti-PD-1 antibody demonstrated improved tumor control at the distant site in comparison with any of the double combinations of the three [49].

A phase 1 study of STINGa (ADU-S100, an aforementioned STINGa) plus PD-1 checkpoint inhibitor (NCT03172936) has been terminated, while another phase 1 study of STINGa (ADU-S100) plus the anti-CTLA-4 antibody ipilimumab (NCT02675439) has been completed for patients with lymphoma and advanced /metastatic solid tumors.

## 9. Agonist Anti–4-1BB/CD137 Antibody Monotherapy

### 9.1. Anti–4-1BB/CD137 Antibody Monotherapy

4-1BB/CD137 is a costimulatory member of the TNF receptor superfamily that is expressed on multiple immune cells following activation, such as T cells, DCs, and NK cells. The long-lasting antitumor activity of agonist MoAbs to 4-1BB/CD137 was principally attributed to CD8^+^ T cells associated with memory responses [182]. Segal and Levy et al. reported the results of an integrated analysis of three trials of the agonist IgG4 anti-4-1BB/CD137 MoAb urelumab [46]. They analyzed the safety profile of urelumab from the data of 346 patients enrolled into three trials including CA186-011, a phase 1 trial that included dose escalation (part 1) employing a 6 + 9 design, cohort expansion (part 2), and additional tumor-specific cohorts (part 3) of 10 patients each with B-cell NHL, colorectal cancer and head neck cancer [182]. In the study CA186-011 [183], urelumab was administered at 0.1 mg/kg every 3 weeks and was adopted in part 1 and 0.3 mg/kg every 3 weeks was performed in part 3. The most common treatment-related AEs were increased aspartate aminotransferase (AST) and alanine aminotransferase (27% each), and fatigue (24%) at doses ≥1 mg/kg, whereas increased AST and fatigue (14% each) was at 0.3 mg/kg, and fatigue (16%) and nausea (13%) at 0.1 mg/kg. Treatment-related AEs of ≥G3 were more frequently observed at doses of 1 to 15 mg/kg than at the 0.1 and 0.3 mg/kg doses. Laboratory data displayed a decrease in absolute neutrophil, platelet, or leukocyte counts of G1-4 across the dose range evaluated. The expression of several IFN-response genes was upregulated in whole-blood samples at approximately 3 and 7 days after administration and returned to baseline levels by day 22. Transaminitis was more frequent at urelumab doses ≥ 1 mg/kg. Indeed, this was the critical factor leading to the development of transaminitis. One patient treated with 6 mg/kg urelumab who experienced severe transaminase elevations underwent a liver biopsy, and histopathological examination revealed a modest mixed inflammatory infiltrate including lymphocytes and other inflammatory cells containing neutrophils. Liver lymphocyte infiltration was observed in mice treated with mouse anti-CD137 MoAb. Collectively, liver-related AEs seemed to be associated with the agonist CD137 MoAbs [46].

### 9.2. 4-1BB/CD137 Agonist with Rituximab

Utomilumab (PF-05082566) is a fully human IgG2 4-1BB/CD137 agonist MoAb causing nuclear factor-kappa B (NF-κB) activation and downstream cytokine production in cell lines and primary lymphocytes [184]. In the dose-escalation cohort, 67 patients were treated with utomilumab at 0.03 to 10.0 mg/kg every 4 weeks and 375 mg/m^2^ rituximab weekly, and in the dose-expansion cohort, utomilumab at 1.2 mg/kg every 4 weeks with the same dose and schedule of rituximab [47]. Seventy percent (47/67) of the enrolled patients had FL and others had CD20^+^ R/R NHL, including 10% DLBCL (7/67). Any of the dose-limiting toxicities were observed, and the MTD for utomilumab combined with rituximab was not reached and estimated to be ≥ 10.0 mg/kg every 4 weeks. All CRs and most PRs were achieved by patients enrolled at dose levels of ≤1.2 mg/kg. The ORR was 21% in all patients with NHL, including 4 CRs and 10 PRs. Median duration of response was 20 months, and median PFS for all treated patients was 4.6 months. No relationship was observed between CD8^+^ T-cell expansion and tumor response. Among the expression of PD-L1, CD8, FOXP3, phospho-STAT33, Granzyme B, and Perforin in tumors at baseline examined by immunohistochemistry, only the elevated expression of FOXP3 was significantly associated with OR, whereas similar trends were found for the other biomarkers. There were significant correlations between gene signatures for lymphocytes, T cells, and CD8^+^ T cells identified by flow cytometry and immune cell fractions derived from deconvolution executed for RNA-sequencing analysis in PB samples at baseline from FL patients. Gene set enrichment analysis discovered gene sets characteristic of CD4^+^ and CD8^+^ T-cell subsets that were related to inferior PFS. As clinical benefit was observed in some patients even with immunosuppressive biomarkers including PD-L1 and FOXP3, by evoking NK-cell activation, 4-1BB/CD137 agonist may have potential to overcome adaptive T cell resistance induced by antitumor response in patients with the tumors that progress on therapy despite of infiltrating T cells [47], immunologically so called “hot” tumors. 

Thus, 4-1BB/CD137 agonist MoAb is a candidate drug that could be combined with CAR T-cell therapy. In fact, a phase 1/2 trial (ZUMA-11 trial [NCT03704298]) of axicabtagene ciloleucel [15], which contains the CD28 co-stimulatory molecule domain, combined with utomilumab, has been performrd for patients with refractory large B-cell lymphoma.

## 10. OX40/CD134 Agonistic MoAb with Rituximab

OX40/CD134 is a type I transmembrane glycoprotein that is transiently expressed on activated T cells, NK cells, NKT cells, and neutrophils [185,186]. OX40/CD134 is constitutively expressed on Tregs in mice, but only upon activation on human Tregs [185]. CD134 ligand (CD134L)/CD252 or agonistic MoAb bound to CD134 results in the recruitment of adaptor proteins called TNF receptor associated factors and stimulation of NF-κB [187,188], phosphatidylinositol 3-kinase/protein kinase B [189], and nuclear factor of activated T-cell [190] pathways. The anti-tumor efficacy of OX40/CD134 has been attributed to IT Treg depletion or inactivation [191,192] and CD4 and/or CD8 stimulation [193,194,195]. In a clinical trial of a murine IgG1 anti- OX40/CD134 MoAb in patients with late-stage cancers, transient expansion of effector CD4^+^ T, CD8^+^ T, and NK cells was observed in some patients who showed tumor regression [196]. CD134L-expressing pDCs bound to CD134 on NK cells led to the release of IFN-γ in CD134L-deficient mice. OX40/CD134 is upregulated in activated NK cells in mice and humans. However, in humans, the overexpression of OX40/CD134 on NK cells was transient and requires close contact with activated T cells or monocytes. Stimulation of OX40/CD134 with an agonistic MoAb augments the therapeutic efficacy of anti-CD20 treatment in a B-cell lymphoma mouse model and in an NK-dependent fashion. Engagement of OX40/CD134 on human NK cells with an agonistic MoAb augmented ADCC capacity and IFN-γ secretion and showed a trend towards greater TNFα release. The functional effects on NK cells are rigorous and dependent on the concomitant FcγR cross-linking [48].

As aforementioned in Section 5, a phase 1 study of IT SD-101 in combination with anti-OX40 antibody BMS 986178 and local low-dose irradiation is ongoing for patients with low-grade B-cell NHL (NCT03410901).

## 11. Conclusions

Lymphoma cells are surrounded by numerous immune cells in LNs and sometimes in the BM, which may be a different environment in the case of leukemia. However, a limitation of this research field is the inherent difference in the cytokine and chemokine milieu between mice and humans. In spite of this, in the FL microenvironment, follicular helper T cells and T-follicular regulatory cells, which support tumor growth, may be mitigated by new agents, such as idelalisib and tazemetostat [143]. On the other hand, adoptive T-cell therapy has come into the limelight within the clinical lymphoma field, although, in lymphoma cases, recruitment of adoptive T cells in LNs and their persistence in patient bodies are obstacles that remain to be solved. Indeed, agents that activate innate immunity per se had not been powerful in the era before adaptive T-cell immunotherapy emerged. In the microenvironment modified by the aforementioned therapeutic agents, innate immune-based therapy will play a pivotal role to improve its efficacy by combining with the new T-cell therapies including CAR T-cell therapy and bispecific T-cell engagers. On the other hand, the required dose of CAR T cells can be reduced in combination, thereby diminished the associated toxicities, such as cytokine-release syndrome and neurotoxicity. Moreover, macrophages serve as APCs in addition to their tumoricidal properties. Therefore, enhanced phagocytosis by anti-CD47 antibodies can lead to enhanced adaptive IR. Anti-CD47 antibody therapy has the potency to raise a persistent antitumor response by acting at the interface of the innate and adaptive immune systems. Agonists of innate immune pathways including TLR9 and STING should be considered to activate DCs and originate effective priming of T cells to steer forward through the cancer-immunity cycle. Additionally, therapeutic strategies that promote adequate inflammation or innate immune activation in the tumor microenvironment should be considered to facilitate the persistence of adaptive immunity. It is possible that radiation therapy, certain chemotherapeutic agents, and B-cell receptor signaling pathway inhibitors achieve this effect by inducing ICD. 

## Figures and Tables

**Figure 1 cancers-14-00141-f001:**
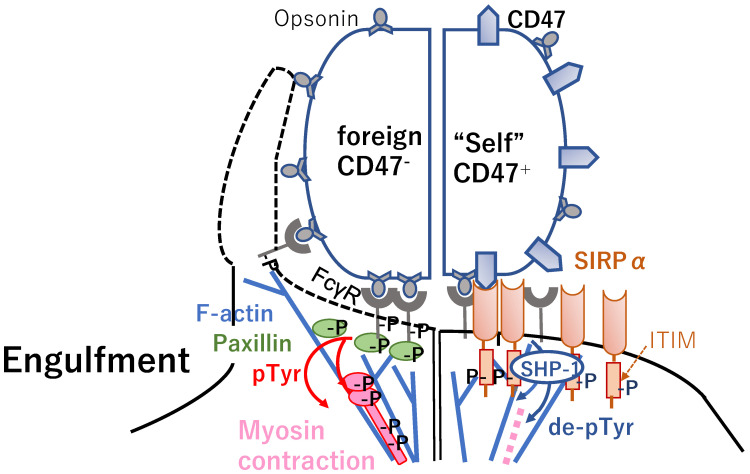
Engagement of CD47 (i.e.,”don’t eat me” signal) on lymphoma cells with SIRPα on macrophages causes activation and phosphorylation of SIRPα ITIM motifs and the recruitment of SHP-1 phosphatases (**right**) preventing myosin-IIA accumulation at the phagocytic synapse (**left**), which results in inhibiting lymphoma cell phagocytosis. IgG activates the FcγR receptor on the phagocyte and induces actin cytoskeleton assembly. IgG-opsonized target cells lacking CD47 results in binding FcγR, leading to activated assembly of paxillin, F-actin, and non-muscle myosin IIA at the phagocytic synapse (**left**). This F-actin assembly is independent of the signaling pathway of CD47. In contrast, parallel interactions with CD47 signals through SIRPα to inhibit myosin assembly and contraction contribute to efficient phagocytosis (**right**). αCD47 Abs, antiCD47 antibodies; APCs, antigen-presenting cells; FcγR, Fcγ receptor; IgG, immunoglobulin G; ITIM, immunoreceptor tyrosine-based inhibitory motif; MØ, macrophage; MHC, major histocompatibility complex; pTyr, phosphotyrosine; SHP-1, Src-homology 2 domain-containing protein tyrosine phosphatase-1; SIRPα, signal regulatory protein alpha.

**Figure 2 cancers-14-00141-f002:**
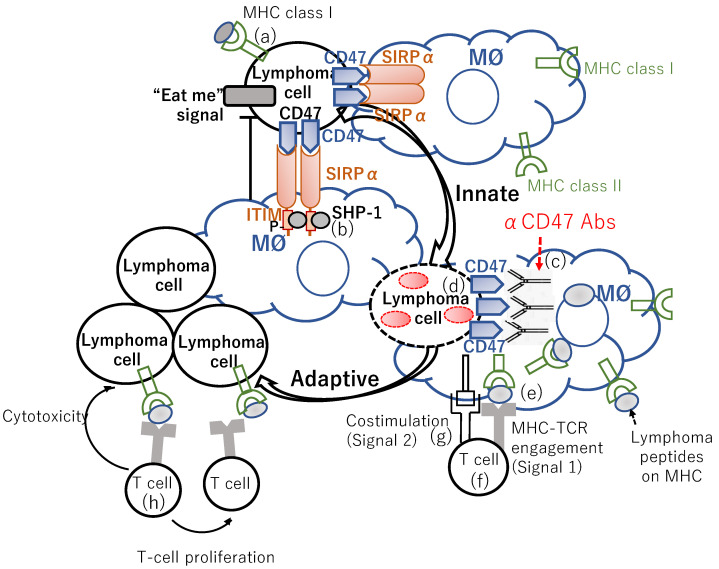
Lymphoma cells express MHC class I (**a**), surface markers of “self,” antiphagocytic (“don’t eat me,”) and phagocytic (“eat me”) signals. Engagement of CD47 (i.e., “don’t eat me” signal) on lymphoma cells with SIRPα on macrophages causes activation and phosphorylation of SIRPα ITIM motifs and the recruitment of SHP-1 phosphatases (**b**) preventing myosin-IIA accumulation at the phagocytic synapse, which results in inhibiting lymphoma cell phagocytosis. Blocking the CD47:SIRPα engagement with anti-CD47 antibodies (**c**) leads to an increase in lymphoma cell phagocytosis by APCs. In turn, the engulfed lymphoma cells are processed (**d**), and these APCs present lymphoma-associated antigens on their MHC (**e**). Then, naïve tumor reactive T cells (**f**) can engage with MHC on the APCs that present lymphoma neoantigens with additional costimulatory molecules (**g**). These lymphoma-specific T cells are thereby activated, expand, and can cause antigen-specific lymphoma cell cytotoxicity (**h**) on remaining lymphoma cells. TCR, T-cell receptor.

**Figure 3 cancers-14-00141-f003:**
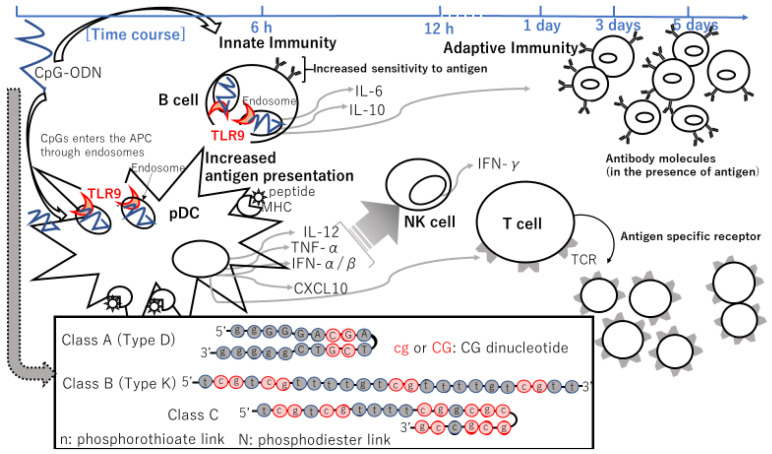
CpG-ODN cellular mechanism of action. DNA containing one or more CpG motifs is taken up by endocytosis in most cell types, but only activates cells expressing the TLR9 receptor (B cells and pDC in humans). These cells create a T_H_1-like cytokine milieu by secreting IFN-α/β, IL-12, CXCL10, and other Th1-promoting cytokines and chemokines. NK cells are secondarily activated, secreting IFN-γ and gaining lytic activity. In addition, B cells become more sensitive to activation through their antigen receptors, and both B cells and pDCs have enhanced expression of costimulatory molecules, improving their ability to activate T-cell responses. Class A (also called Type D) ODNs have a phosphodiester core flanked by phosphorothioate terminal nucleotides (in **box**). They carry a single CpG (indicated in red circles) motif flanked by a palindromic sequence that enables the formation of a stem-loop structure. Class A ODN also possesses poly G motifs at the 3′ and 5′ ends that promote concatamer formation. Class A ODNs initiate pDC to mature and secrete IFN-α but have no effect on B cells. Class A ODNs are kept for longer periods in the early endosome. Class B (also referred to as Type K) ODNs contain 1–5 CpG motifs (in red circles) typically on a phosphorothioate backbone. Class B ODNs trigger pDC to differentiate and produce TNF-α and stimulate B cells to proliferate and secrete IgM. Class B ODNs are rapidly transported through early endosomes into late endosomes. This backbone augments resistance to nuclease digestion. Class C ODNs resemble Class B ODNs in being comprised completely of phosphorothioate nucleotides but are similar to Class A ODNs in including palindromic CpG motifs that can form stem-loop conformation or dimers. Class C ODNs provoke B cells to produce IL-6 and pDCs to secrete IFN-α. Class C ODNs keep activity in both early and late endosomes, and therefore have characteristics in common with both Class A and Class B ODNs. APC, antigen-presenting cell; CpG-ODNs, CpG oligodeoxynucleotides; CXCL10, C-X-C motif chemokine ligand 10; IP-10, IFN-gamma-inducible protein of 10 kilodalton; pDCs, plasmacytoid dendritic cells; MHC, major histocompatibility complex; NK, natural killer; TCR, T-cell receptor; TLR9, Toll-like receptor 9.

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
