# Peer review of "Approaches of the Innate Immune System to Ameliorate Adaptive Immunotherapy for B-Cell Non-Hodgkin Lymphoma in Their Microenvironment"

_cancers, 2021, doi:10.3390/cancers14010141_

Round 1

Reviewer 1 Report

The paper by Watanabe is about the role of innate immune system  to Ameliorate Adaptive Immunotherapy for B-Cell Non-Hodgkin Lymphoma in Their Microenvironment. The paper finds merit and this field is intriguing. However, some issue have to be addressed before the paper could be accepted.

  1. Introduction. New therapies such as lenalidomide, idelalisib, ibrutinib for NHL should be briefly discussed. The authors adfirm FL can only be cured with allogeneic transplantation, it is true but it is too pessimistic, FL patients have a long survival and could be successfully managed with new therapies, even if relapsed/refractory. Allotransplant is rarely performed, please reformulate this paragraph. Even for MCL, new strategies (rituximab maintenance, ibrutinib) have prolonged PFS and OS in R/R patients and should be cited. The authors should speak more widely about CAR-T cell therapy for DLBCL and should remove the CARVac treatment that is about solid tumors (ref.17). In the introduction you should specify that STINGa induce response mainly in preclinical models. In the Introduction and/or in the paragraph 2. you should briefly describe TAM M1 and M2 (see MAntovani et al. NAt rev clin Oncol; Cencini et al. Cancers 2021).
  2. You should describe possible other TAM-based therapeutic approaches, such as Macrophage killing due to CCL2-CCR2
    signalling axis inhibition (with trabectedin), antiangiogenic (lenalidomide), CSF-1R signalling inhibitors, dacetuzumab, TAM reprogramming with artesunate (maybe also in a table).
  3. You should add the year of the cited paper(2002) at the beginning. The paragraph, especially the pre-clinical part should be shortened.
  4. 4.1The paragraph should be shortened, even because the figure 3 legend is very long. The Figure 2 is maybe figure 3 at page 9? PAragraph 4.2 many papers were published more than 10 years ago,a re there more recent papers? If not, try to explain why, what are the difficulties that limit the applicability of research in clinical practice. Paragraph 4.4. Please specify ibrutinib dosage and days of administration. Paragraph 4.5 Results are comparable to those achieved by RT alone. You should briefly cyte data about RT alone for R/R FL cases. PAge 18. You say "vaccination might be more effective in previously untreated patients especially in case of prior anti-CD20 antibodies". If untreated, patients have not received anti-CD20 antibodies, please clarify. You could cite some clinical experience about anti CD20 and RT for FL cases and describe the rationale of abscopal effect. What is Ann Arbor stage, if available, in patients enrolled in the study by Frank and Levy?

Table 1. You should remove the paper about solid tumors. You should add i nthe efficacy column the duration of response and survival data, if available.

         5. Due to the scarcity of data about NHL, the paragraph should be shortened. 

6. Page 19. What is the aim of second paragraph "in patients who achieved PR? It is not clear, please reformulate or clarify. End of page ELISOPT is maybe ELISPOT.

8. You should specify future perspectives about NHL, combination therapies? Are there ongoing clinical trials?

10. Add a paragraph about new perspectives, strenght and limitations of this research field. What is the position of innate immune-based therapy in the therapeutic scenario in the era of new drugs and CAR-T?

Author Response

I have added the following Word file.

Reviewer 2 Report

The paper is well written and provides relevant information on an emerging field of considerable interest.   

Minor revisions:

- Consider reducing references to solid tumors to make the work more concise and easily readable.

-  (line 132) The sentence is grammatically incorrect and must be rewritten.

Author Response

  1. Following references concerning solid tumors have been deleted and the contents citing these references have been deleted to shorten the text: References 17, 96 to103, 122, 123, 153, and 176.

  1. The sentence in line 132 has been corrected as follows: “In contrast, a multicenter, open-label, first-in-human phase 1 study of TTI-621 was conducted, wherein a recombinant soluble fusion protein composed of the CD47-binding domain of human SIRPα and the Fc region of human IgG1 binds to CD47 in patients with R/R hematologic malignancies.”

Reviewer 3 Report

The review is comprehensive and well organized, but the multiple grammatical errors and poor writing make it difficult to correctly assess the work, so extensive editing of English language and style is required for its subsequent scientific assessment. Readers would find it helpful to include at the end of each section a summary of the achievements of each treatment modality and future perspectives, including active clinical trials that include any of these therapies.

Author Response

At the end of each Section, a summary of the representative achievement of Section or/and future perspective or active clinical trials with ClicalTrials.gov Identifier numbers in parentheses, currently ongoing and of which the recruitment has been just finished and the results are unpublished yet have been added according to Reviewer 3’s comments.

Round 2

Reviewer 1 Report

The paper has been significantly improvved by the author. All issues have been clarified. 

The paper could be accepted in its current form.

Reviewer 3 Report

The manuscript has been sufficiently improved to warrant publication.